# First description of a herpesvirus infection in genus Lepus

F. A. Abade dos Santos[1,2]*, M. Monteiro[1], A. Pinto[3,4], C. L. Carvalho[1], M. C. Peleteiro[2], P. Carvalho[1], P. Mendonça[1], T. Carvalho[3], M. D. Duarte[1,2]

1 Instituto Nacional de Investigação Agrária e Veterinária, Oeiras, Portugal, 2 Centre for Interdisciplinary Research in Animal Health (CIISA), Faculdade de Medicina Veterinária, Universidade de Lisboa, Lisboa, Portugal, 3 Instituto de Medicina Molecular João Lobo Antunes, Faculdade de Medicina, Universidade de Lisboa, Avenida Professor Egas Moniz, Lisboa, Portugal, 4 EM Suite, Royal Brompton Hospital and Harefield NHS Foundation Trust, Lisboa, Portugal

* faas@fmv.ulisboa.pt

**Data Availability Statement:** All the data are on the manuscript.

**Funding:** Most of the field and laboratory work referred to in this manuscript was supported by the Fundação para a Ciência e Tecnologia (FCT) (Grant

## Abstract

During the necropsies of Iberian hares obtained in 2018/2019, along with signs of the nodular form of myxomatosis, other unexpected external lesions were also observed. Histopathology revealed nuclear inclusion bodies in stromal cells suggesting the additional presence of a nuclear replicating virus. Transmission electron microscopy further demonstrated the presence of herpesvirus particles in the tissues of affected hares. We confirmed the presence of herpesvirus in 13 MYXV-positive hares by PCR and sequencing analysis. Herpesvirus-DNA was also detected in seven healthy hares, suggesting its asymptomatic circulation. Phylogenetic analysis based on concatenated partial sequences of DNA polymerase gene and glycoprotein B gene enabled greater resolution than analysing the sequences individually. The hare' virus was classified close to herpesviruses from rodents within the Rhadinovirus genus of the gammaherpesvirus subfamily. We propose to name this new virus Leporid gammaherpesvirus 5 (LeHV-5), according to the International Committee on Taxonomy of Viruses standards. The impact of herpesvirus infection on the reproduction and mortality of the Iberian hare is yet unknown but may aggravate the decline of wild populations caused by the recently emerged natural recombinant myxoma virus.

## 1. Introduction

The Iberian hare (*Lepus granatensis*), also known as Granada hare, is an endemic specie of the Iberian Peninsula whose populations are considered stable by the IUCN holding a '*minor concern*' conservation status [1].

*Lepus granatensis* is the only hare species found in Portugal and the most widespread in the Iberia [2], therefore, highly relevant for biodiversity preservation and hunting activity in both countries, particularly for greyhound racing.

Contrarily to the wild rabbit, which drastic decline has been linked, among other factors, to viral epizooties, until recently, the Iberian hare was not affected by viral diseases. Environmental and anthropogenic factors, however, have had a negative impact on both hare and wild-rabbit populations.

SFRH/BD/137067/2018), Fundo Florestal Permanente (Government of Portugal) in the scope of the Action Plan for the Control of Rabbit Viral Haemorrhagic Disease (+COELHO, Dispatch no. 4757/2017 of 31 May) and by the Centre for Interdisciplinary Research in Animal Health, Faculty of Veterinary Medicine, University of Lisbon (CIISA, FMV-UL) (Portugal) (Project UID/CVT/00276/2013). Funding bodies played no direct role in the design or conclusion of the study.

**Competing interests:** No competing interests.

*Lepus granatensis* was considered naturally resistant to myxomatosis, which is endemic in Iberian Peninsula since 1956 [3], despite very sporadic reports of the disease in the European brown hare (*L. europaeus*), namely in France and Ireland [4–6]. However, during the summer and autumn of 2018 outbreaks of myxomatosis in the Iberian hare were reported in Spain [7] and Portugal [8], respectively.

No other diseases of viral origin have been described in the Iberian hare, including those caused by herpesviruses, which may have a fatal outcome in rabbits [9].

Until now, four herpesviruses have been identified in leporids: Leporid herpesvirus 1 (LeHV-1), Leporid herpesvirus 2 (LeHV-2), Leporid herpesvirus 3 (LeHV-3) and Leporid herpesvirus 4 (LeHV-4) (Table 1). Of these, the most common naturally occurring herpesvirus

**Table 1. Summary of the characteristics of the four herpesviruses identified in leporids.**

| Type | Subfamily | | Common Name | Host | Physiopathology | Isolation |
|---|---|---|---|---|---|---|
| **LeHV-1** | γ | Not attributed[a] | Cottontail herpesvirus | *Sylvilagus floridanus* | | Isolated from primary kidney cells cultures of *Sylvilagus floridanus* [13]. |
| | | | | | | No report of disease in domestic rabbits, since *Oryctolagus cuniculus* is not infected [10]. |
| **LeHV-2** | γ | Not attributed[a] | Herpesvirus cuniculi | *Oryctolagus cuniculus* | Some evidences of a subclinical encephalitis in infected New Zealand white rabbits [30]. | Isolated in 1968 from kidneys of apparently healthy *Sylvilagus floridanus* [31]. |
| | | | | | | *Oryctolagus cuniculus* is the natural host where infection is asymptomatic[32] |
| **LeHV-3** | γ | Not attributed[a] | Herpesvirus sylvilagus | *Sylvilagus floridanus* | Lymphoproliferative disease and tumour-like lesions in the lymph nodes, kidney, spleen, and liver [31,33]. | Isolated from primary kidney cells cultures of Sylvilagus floridanus [13]. |
| | | | | | | *Oryctolagus cuniculus* is not infected |
| | | | | | | Not isolated in WI-38, HeLa, Chang's conjunctiva, human amnion (FL), green monkey kidney (Vero), primary rhesus monkey kidney, primary hamster kidney, BHK-21, primary mouse embryo, and primary chick embryo [33]. Isolated in DRK-3 cells [33]. |
| | | | | | | CPE appear after 10–15 days of inoculation. Infected cells show focal areas of round and distorted cells, and in 1–2 days, emerged syncytial masses containing 50 or more nuclei [31]. |
| | | | | | | H&E coloration show typical type A intranuclear inclusions in the infected cells. Complete cell destruction occurred after a 5 to 7-days period [31]. |
| **LeHV-4** | α | Attributed[b] | | *Oryctolagus cuniculus* | Lethargy, anorexia, conjunctivitis, fever, and abortion. Haemorrhagic dermatitis, splenic necrosis, hepatic necrosis, and multifocal pulmonary haemorrhage and oedema. Distinctive glassy eosinophilic herpetic intranuclear inclusion bodies were observed in the skin fibroblasts? And mesenchymal cells of the spleen and lung [9,34]. | Isolated in rabbit skin (RS), RK13 and Vero cells [9]. CPE characterized by syncytium formation, cell enlargement, and cell lysis, similar to human herpesvirus type 1 (HHV-1). Jin et al. [9] verified that in rabbits inoculated with LHV-4, the appendix, sciatic nerve, kidney, adrenal gland, and many other organs were positive for the virus at the 5-days post infection(dpi), while at the 14 dpi only trigeminal ganglia eye and tonsil were positive. |
| **LeHV-5** | γ | Not attributed | Iberian hare herpesvirus | *Lepus granatensis* | Described in the results of this work | |

[a]Not attributed by the ICTV; Viruses which may be members of the genus Rhadinovirus [35] but have not been approved as species

[b]Leporid alphaherpesvirus //Leporid Herpesvirus 4

infections identified in rabbits are LeHV-2 and LeHV-3 (reviewed by [9]), which alongside LeHV-1 belong to the Gammaherpesvirinae subfamily. Conversely, LeHV-4 is a member of the Alphaherpesvirinae subfamily. These distinct herpesviruses have a broadly variable impact on the European rabbit, with LeHV-2 and LeHV-3 infections usually passing unnoticed [10], while LeHV-4 is far more aggressive, causing fatal infections [9].

Herpesviruses are enveloped viruses, of 200–250 nm in diameter, organised in four concentric layers [11], 1) a core with the linear dsDNA genome, 2) T = 16 icosahedral capsid with about 125nm of diameter surrounded by a 3) proteinaceous tegument that contains many virus-coded proteins and enclosed in a 4) lipid envelope containing several viral glycoproteins. Morphologically, herpesviruses are distinct from all other viruses [12], and therefore easily recognised by electron microscopy.

Herpesviruses belong to order Herpesvirales that comprises three families, namely the Herpesviridae family, which includes more than 100 viruses of mammals, birds and reptiles and whose members have large genomes ranging from 125 to 290kb [13], the Alloherpesviridae family, which includes the fish and frog viruses, and the Malacoherpesviridae family, which contains the bivalve virus [12].

The Herpesviridae family includes the subfamilies Alphaherpesvirinae, Betaherpesvirinae and Gammaherpesvirinae. Their members have different biologic properties and distinct classification, supported by phylogeny. The Gammaherpesvirinae subfamily is divided into four genera, namely *Macavirus*, *Percavirus*, *Lymphocryptovirus*, and *Rhadinovirus* [12].

While the subfamily Alphaherpesvirinae causes rapid lysis in cell culture, members of Betaherpesvirinae grow slowly inducing the formation of giant cells in culture, and Gammaherpesvirinae typically infect lymphoid tissue, meaning a primary tropism for lymphoid lineage cells [14], which can lead to lymphoproliferative diseases [9] and oncogenesis [13].

In this study, we investigated the presence of herpesvirus in myxoma virus (MYXV)-positive hares that alongside, the typical myxoma virus-induced skin lesions, presented other lesions in the genitalia, eyelids, lips and nose suggestive of herpesvirus infection.

To unveil the prevalence of herpesvirus in the hare populations, healthy hares were also investigated.

## 2. Materials and methods

### 2.1. Sample

A total of 38 Iberian hares, none sacrificed for the purpose of this study, were investigated within the scope of a national surveillance program (Dispatch 4757/17, 31th may) [8] in action since August 2017. Of these, 16 were males, 20 were females, and two failed sex-determination. Eighteen hares were legally hunted by the Hunting Associations during the 2018/2019 season (October to December), authorized by permits from the National Forest Authority, the Institute for Nature Conservation and Forests (ICNF), while 20 were found dead in the field between October 2018 and June 2019. None of the authors were responsible for the death of the animals. Hunted and hares found dead were sampled in six Districts of mainland Portugal, namely Setúbal, Santarém, Beja, Évora, Portalegre and Faro, (Fig 1).

### 2.2. Necropsy and histopathology

Cadavers were necropsied and spleen, liver, lung, duodenum and skin samples (namely scrotum, lips and nose) were collected for virology, bacteriology and histopathology. The entire gastrointestinal tract was taken for parasitological analysis. From hunted hares, only spleen, liver and lung samples were received at the laboratory.

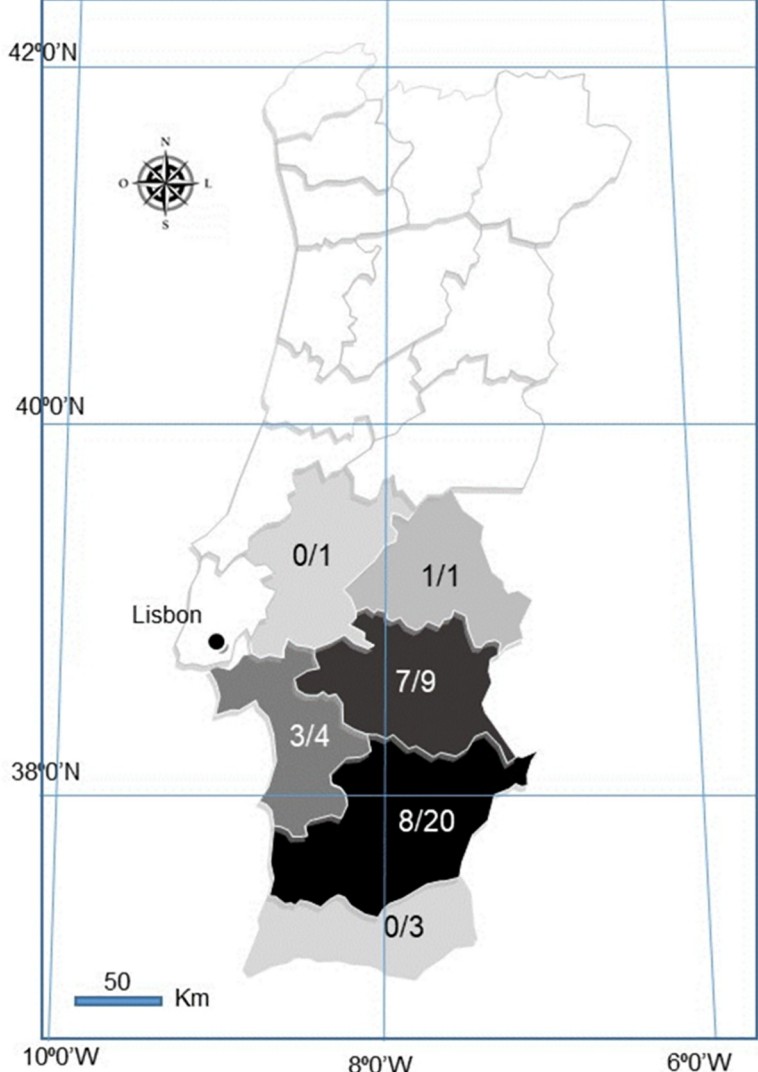

**Fig 1. Map of Portugal showing the geographic origin of the 38 LeHV-5 positive hares over the total sampling per district.** White coloured Districts were not sampled. Darker shades correspond to higher positivity.

For histopathology, skin and genitalia fragments were fixated in 10% neutral buffered formalin, routinely paraffin embedded, sectioned at 4 μm, and stained with Hematoxylin and Eosin (H&E).

## 2.3. Transmission electron microscopy

The fragments selected for transmission electron microscopy (TEM) were formalin fixated for 48h or on a solution 0.1M sodium cacodylate (Sigma©) containing 2.5% gluteraldehyde (Sigma©) for 72h. When the samples were already embebed in paraffin, the regions of interest were extracted from the block, sliced smaller than ~1mm$^3$ with a scalpel blade into two separate viles, and washed thoroughly in xylene. After rehydration using decreasing concentrations of ethanol, fragments were washed in 0.1M cacodylate buffer [15]. Samples were then postfixed with 2% osmium tetroxide (EMS) for 30min, and stained in block with 1% Millipore-filtered uranyl acetate (Agar Scientifics), after which they were dehydrated in increasing

concentrations of ethanol, infiltrated and embedded in EMBed-812 hard (EMS). Polymerization was performed at 60°C for 2 days. Ultrathin sections were cut either in a UC7 ultramicrotome or in a Reichert ultracut E ultramicrotome (Leica), collected to 1% formvar coated copper slot grids (Agar scientific), stained with uranyl acetate and lead citrate (Sigma) and examined in a H-7000 transmission electron microscope (Hitachi) at an accelerating voltage of 100 kV or Jeol 1400plus transmission electron microscope at an accelerating voltage of 120 kV. Digital images were obtained using a Megaview mid mount digital camera (Olympus) or using a AMT XR16 bottom mid mount digital camera (AMT©). The sections were systematically analysed using AMT© software and several high and low magnifications were acquired.

## 2.4. Bacteriological and parasitological examination

Liver, spleen and lung samples were analysed using standard bacteriological methods. Parasitological examination of the intestine was carried out resourcing to direct wet mount, sedimentation and filtration techniques.

To investigate the presence of Enterobacteriacea the ID 32E (Biomerieux®) was used, while for non-Enterobacteriaceaethe API 20NE kit (Biomerieux®) was utilised. To test the presence of Streptococcus and Staphylococcus, the ID 32 STREPT (Biomerieux®) and the ID 32 STAPH kits (Biomerieux®) were used, respectively. The API CORYNE (Biomerieux®) kit was utilized for the identification of Corynebacteria and coryne-like organisms. To investigate the presence of Salmonella, peptone water and Rappaport Vassiliadis semi solid culture media were used. The agarose SMID2 and XLD culture media were utilized, whenever there was a suspicion of Salmonella. Other culture media used for bacterial identification in the samples included the MacConkey agar and the Blood agar culture media.

## 2.5. Virological examination

For nucleic acid extraction, fresh samples of liver and spleen were homogenised at 20% with phosphate buffered saline (PBS) and clarified at 3000g for 5 min. Total DNA and RNA were extracted from 200µl of the clarified supernatants, using the MagAttract 96 cador Pathogen Kit in a BioSprint 96 nucleic acid extractor (Qiagen, Hilden, Germany), according to the manufacturer's protocol.

All the animals were tested for rabbit haemorrhagic disease virus 2 (RHDV2) and MYXV by real time PCR systems described by Duarte *et al* (2015) [12] and Duarte *et al* (2014) [22], respectively. The presence of LEHV-4 was investigated by using the PCR described by Jin *et al* (2008) [9]. A generalist nested PCR directed to the herpesviral *DNA polymerase* that allows the detection of herpesviruses of different subfamilies by Van Devanter *et al.* (1996) [16] was also used.

The glycoprotein B gene was partially amplified using the GH1 system described previously [17].

Amplifications were carried out in a Bio-Rad CFX96™ Thermal Cycler (Bio-Rad Laboratories Srl, Redmond, USA), using the One Step RT-PCR kit (Qiagen, Hilden, Germany) for RHDV2, and the HighFidelity PCR Master Mix (Roche Diagnostics GmbH, Mannheim, Germany), for MYXV and herpesvirus detection, respectively.

Information regarding these methods is summarized in Table 2.

## 2.6. Sequencing analysis

The PCR products were visualised in 2% horizontal electrophoresis agarose gel, purified using the NZYGelpure kit (NZYTECH), and directly sequenced using the ABI Prism BigDye Terminator v3.1 Cycle sequencing kit on a 3130 Genetic Analyser (Applied Biosystems, Foster City,

**Table 2. Molecular methods used for the detection of RHDV2 (Duarte et al, 2015), MYXV (Duarte et al, 2014), LeHV 4 [9] and a wide variety of herpesviral genomes from human and animal herpesviruses (Van Devanter, 1996).**

| Primer | Sequence (5'-3') | Target gene | Amplicon size (bp) | Amplification conditions | Reference |
|---|---|---|---|---|---|
| RHDV2-F | TGGAACTTGGCTTGAGTGTTGA | Vp60 | 127 | 50˚C -45min 95˚C-15min 50x (95˚C-15s 60˚C-30s 72˚C-30s) | [36] |
| RHDV2-R | ACAAGCGTGCTTGTGGACGG | | | | |
| RHDV2-Probe | [FAM]-TGTCAGAACTTGTTGACATCCGCCC-[TAMRA] | | | | |
| M000.5R/L-F | CGACGTAGATTTATCGTATACC | M000.5 | 125 | 95˚C- 10min 45X (95˚C-30s 50ºC-30s 60˚C-30s) | [37] |
| M000.5R/L-R | GTCTGTCTATGTATTCTATCTCC4 | | | | |
| MYXV-probe | [FAM]-TCTATGTCTGCCCGAGGATAGA-[TAMRA] | | | | |
| LeHV-4 F1F | ATGACGCCCACCAACGTCTC GCACAGTGTGTGTTAGACG TGTGGCCAAGAACAACGATA | | 1617 1162 945 | 95ºC-10min 35x (95º-15s 54ºC-20s 72ºC-3min) 72ºC-10min | [34] |
| LeHV-4 F2F | | | | | |
| LeHV-4 F3F | | | | | |
| LeHV-4 F1R | CATAGACCGTAGGCGGTTC | | | | |
| LeHV-4 F2R | ACGTGAACAGGAACCGGTAG | | | | |
| LeHV-4 F3R | CTAGAGGTCGTTCACCACCG | | | | |
| DFA (F 1st round) | GAYT TYGCNAGYYTNTAYCC | DNA polymerase | 215 to 315 | 94ºC- 10min 35x (94ºC-60s 46ºC-60s 72ºC-3min) | [16] |
| ILT (F1st round) | TCCTGGACAAGCAGCARNYS GCNMTNAA | | | | |
| TGV (F1st round) | TGTAACTCGGTGTAYGGNTTYACNGGNGT | | | | |
| KG1 (R 2nd round) | GTCTTGCTCACC AGNTCNACNCCYTT | | | | |
| IYG (R 2nd round) | CACAGAGTCCGTRTCNCCRTADAT | | | | |
| 2759s | CCTCCCAGGTTCARTWYGCMT AYGA | gB | 700 bp | 94ºC- 10min 35x (94ºC-60s 46ºC-60s 72ºC-3min) | [17] |
| 2762as | CCGTTGAGGTTCTGAGTGTAR TARTTRTAYTC | | | | |
| 2760s | AAGATCAACCCCAC(N/I)AG (N/I)GT(N/I)ATG | | 500bp | | |
| 2761as | GTGTAGTAGTTGTACTCCCTR AACAT(N/I)GTYTC | | | | |

CA, U.S.A). Nucleotide sequences were analysed and assembled into consensus sequences using the BioEdit version 7.2.5 software, and submitted to GenBank. Nucleotide sequences were translated using Mega X 10.1 software.

## 2.7. Phylogenetic analysis

Partial nucleotide (171bp) sequences of the viral *DNA polymerase* gene were aligned using the Clustal W with gap opening penalty and a gap extend penalty of 30 and 15, respectively. A phylogenetic analysis was conducted in MEGA X [18], using the model selected by Model function (MEGA X).

The evolutionary history of 28 partial DNA polymerase protein sequences of gammaherpesviruses was inferred by Maximum Likelihood. The Hasegawa-Kishino-Yano (HKY) model [19], which showed the lowest Bayesian Information Criterion (BIC) and Akaike Information Criterion corrected (AICc) values was used. A discrete Gamma distribution (G) was used to model evolutionary rate differences among sites. The rate variation model allowed for some sites to be evolutionarily invariable (I).

For more accurate phylogenetic analysis of the hare herpesvirus described is this manuscript, the partial nucleotide (171bp) sequences of the viral DNA polymerase gene catalytic subunit was concatenated with the partial (453bp) nucleotide sequences of the viral Glycoprotein B gene belonging to the same strain. The sequences were translated and aligned using the Clustal W with gap opening penalty and a gap extend penalty of 30 and 15, respectively. The

final alignment was edited to include all the sequences, corresponding to 570 nucleotides and 190 amino acids of length.

The Le Gascuel model (2008) [20], Gamma distributed with invariant sites (LG+G+I) was selected for the protein-based trees, according to BIC (14177.4) and AICC (13542.9) criteria. The analysis involved 45 amino acid sequences.

### 2.8. Herpesvirus isolation

Isolation of herpesvirus was attempted from organs of hares coinfected with MYXV and LeHV-5, namely from liver and spleen, penile and vulva samples. In addition, liver and spleen samples from two hares with single herpesvirus infection, were also used.

Samples were homogenized at 20% in phosphate-buffered saline containing penicillin, streptomycin and amphotericin B (antibiotic-antimycotic), used according to the manufacturer (Gibco, Life Technologies Corporation). Following centrifugation, the supernatant was filtered through a 0.45-μm-pore-size filter (Millipore Express) and used to inoculate sub 70% confluent Candrel R Feline Kidney (CRFK) epithelial cells (ATCC-CCL-94), Vero cells (ATCC No. CRL-1986), Rabbit Kidney (RK13) cells (ATCC-CCL-37) and Hella cells (ATCCNo. CRM-CCL-2)- RK13 cells grown in Eagle medium and the others in Dulbecco's modified Eagle's Media was supplemented with 10% foetal calf serum (Gibco), penicillin, streptomycin and amphotericin B (antibiotic-antimycotic used at 1:100), 50μg/ml gentamicin (Gibco). Cells were maintained at 37˚C under humidified atmosphere with 5% $CO_2$ and observed daily for cytopathic effect (CPE) by phase contrast microscopy. Three passages were carried out. The supernatant and cell pellet of each passage were tested for the presence of herpesvirus by PCR [17].

## 3. Results

### 3.1. Necropsy showed lesions compatible with herpesvirus infection in Iberian hares with myxomatosis

Overall, MYXV-positive hares revealed good body condition and, alongside typical myxomatosis lesions, necrosis of the genitalia was noticed in more than 70% of the hares studied. This lesion was more evident in males, affecting the penile glans, but was also observed in females inthe vulva. Other lesions observed in these hares included the presence of herpetic-like skin vesicles, uncommon in rabbits with myxomatosis.

Further investigation of the macroscopic lesions and histopathological patterns was carried out in hares co-infected with LeHV-5 and MYXV. At this time, we disclose the macroscopic and histopathological findings from two male hares found dead in November 2018 (#38455/18, hereafter designated hare-1) and August 2019 (#25456/19, hereafter designated hare-2).

Hare-1 presented with eyelids thickened by the accumulation of mucopurulent exudate and marked enlargement of the penis measuring 1.3x1cm in diameter (normal diameter is less than 0.5cm) and irregular surface (Fig 2 and 2A) lined with light-yellow dry exudate.

At the necropsy, hare-2 showed ulcerated multinodular thickening of the eyelids and lips. Accumulation of mucopurulent exudate in both eyes was also registered and a small vesicle was present in the lower lip (Fig 3).

### 3.2. Histopathology

The dermis of hare-1 showed fusiform or stellate mesenchymal cell proliferation, surrounded by abundant extracellular matrix, scattered infiltration by lymphocytes and macrophages, and small aggregates of heterophiles, consistent with myxomatosis.

The penile epithelium of this hare was mostly necrotic and replaced by a thick band of necrotic cells, heterophils and red blood cells (Fig 2).

Severe heterophile infiltrations of the stroma, in either a diffuse pattern or multifocal aggregates, were also seen. In the stroma, there was also proliferation of pleomorphic spindle cells, with some nuclei almost filled with slightly eosinophilic inclusion bodies (Cowdry type A inclusions) (Fig 4), suggesting a nuclear replicating virus. These lesions, unexpected in myxomatosis, are compatible with herpesvirus.

In the skin of hare-2, a ballooning degeneration of keratinocytes was registered. Coalescent intra-epidermal and subepidermal vesicopustules (Fig 5) filled with fibrin and necrotic cells debris and multifocal detachment of the eyelids, lips and foreskin epidermis were seen. In the underlying dermis, multifocal haemorrhages, intense infiltration by heterophils and necrotic cells with accumulation of chromatin debris were present (Fig 5).

Below the dermis, accumulation of myxoid tissue with pleomorphic spindle cells, some of which showing rounded or oval and slightly eosinophilic intranuclear inclusion bodies, was observed (Fig 6). An infiltrate of mononucleated inflammatory cells and heterophils was present in skeletal muscle tissue.

### 3.3. Electron microscopy

Samples from hare-1 and hare-2 were further processed and analysed for TEM allowing the confirmation of the presence of herpesvirus in different tissues.

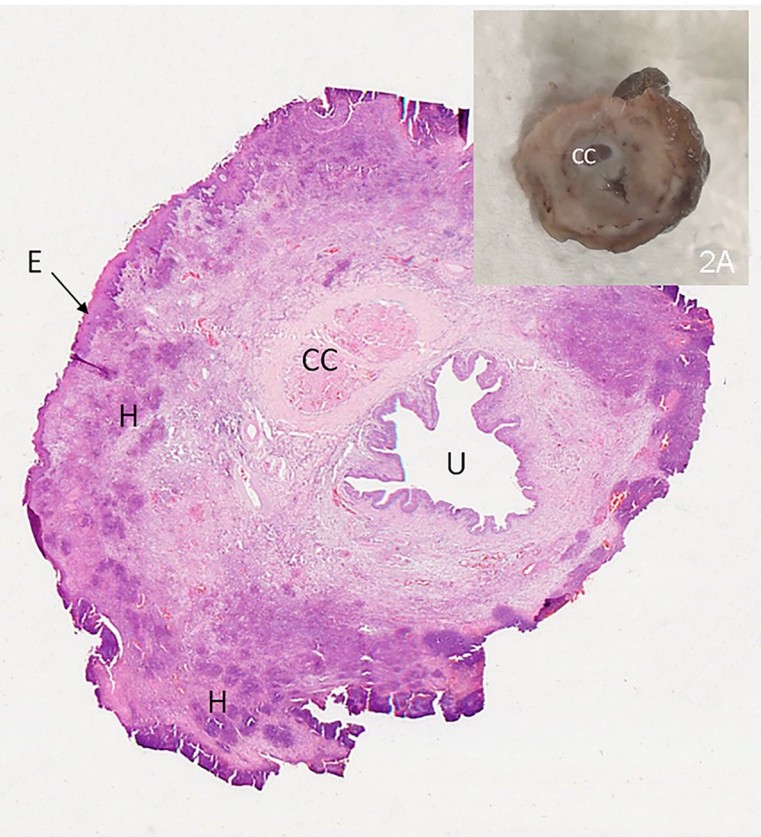

**Fig 2. Penis of hare-1.** 2- H&E staining showing several necrotic areas in the epithelium (E) and multifocal heterophils aggregates in the stromal tissue (H). C*orpus cavernosum* (CC); penile uretra (U). 4x. 2A - Cross section of penis after fixation- exuberant thickening of the penis.

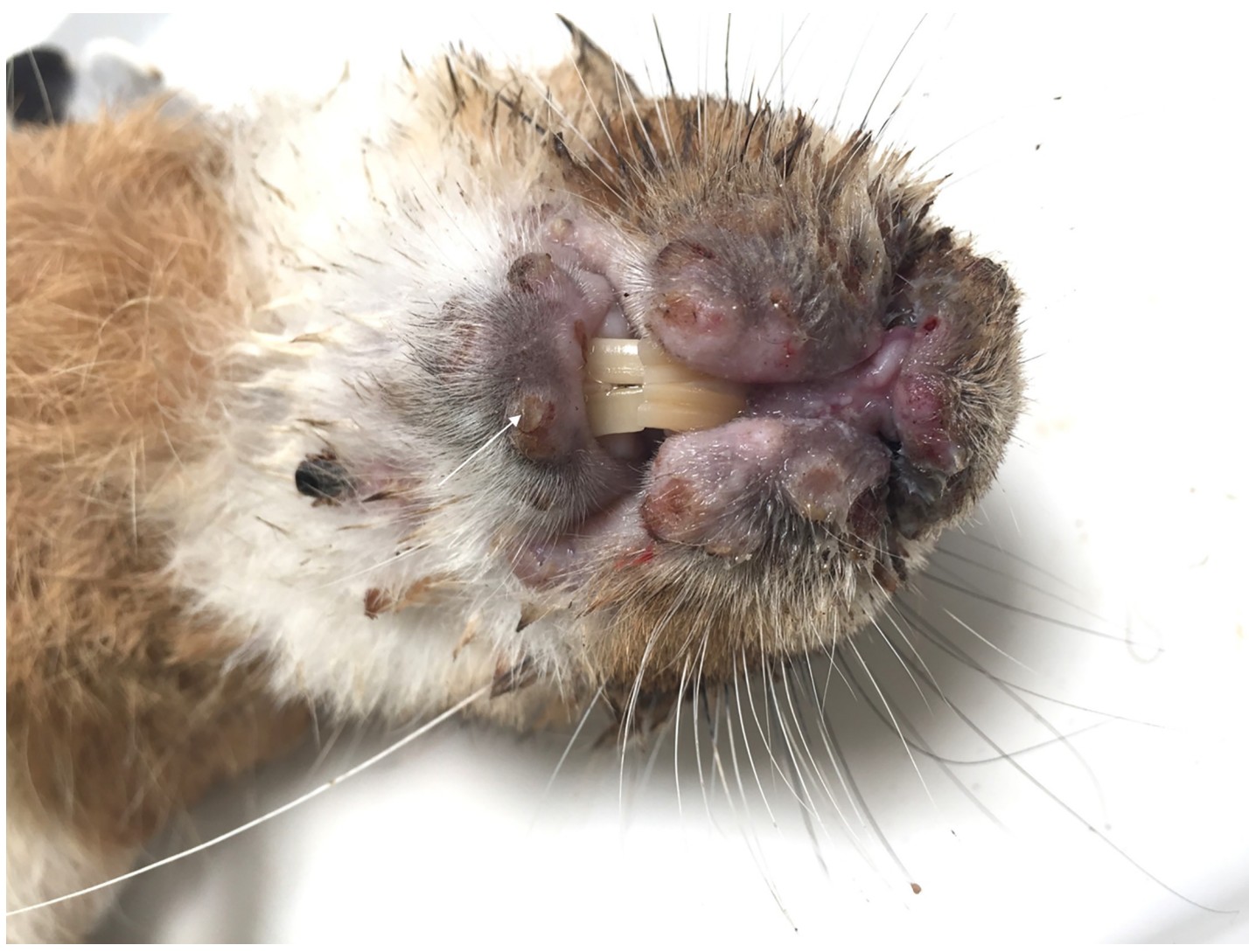

**Fig 3. Hare-2 –Oedema of lips and nose with ulcerated nodules in the upper lip.** A vesicle can be seen in the lower lip (arrow).

In the penile soft tissue of hare-1, spherical virions, with structure and dimensions compatible with herpesviruses, comprising an inner core packed into an icosahedral capsid, were observed in the nucleus of stromal cells (Fig 7), indicating nuclear replication (Fig 7B), which is an attribute of herpesviruses. The viral capsid contained a relatively small, asymmetrical, electron-dense region that probably represents the condensed DNA core. In this animal, also positive to myxomatosis, no MYXV particles were found in the samples processed.

### 3.4. Virological, bacteriological and parasitological results

None of the 38 hares investigated in this study tested positive to RHDV, RHDV2 or LEHV-4. Fifty percent of the hares were positive to LeHV-5, of which 68.4% (13/19) were also positive to MYXV.

Herpesvirus-DNA was also detected by PCR in the liver, spleen and lung samples of 41.2% (7/17) of the apparently healthy hunted hares that tested negative for MYXV. From this group of hares, no genitalia/skin samples were available for histopathology.

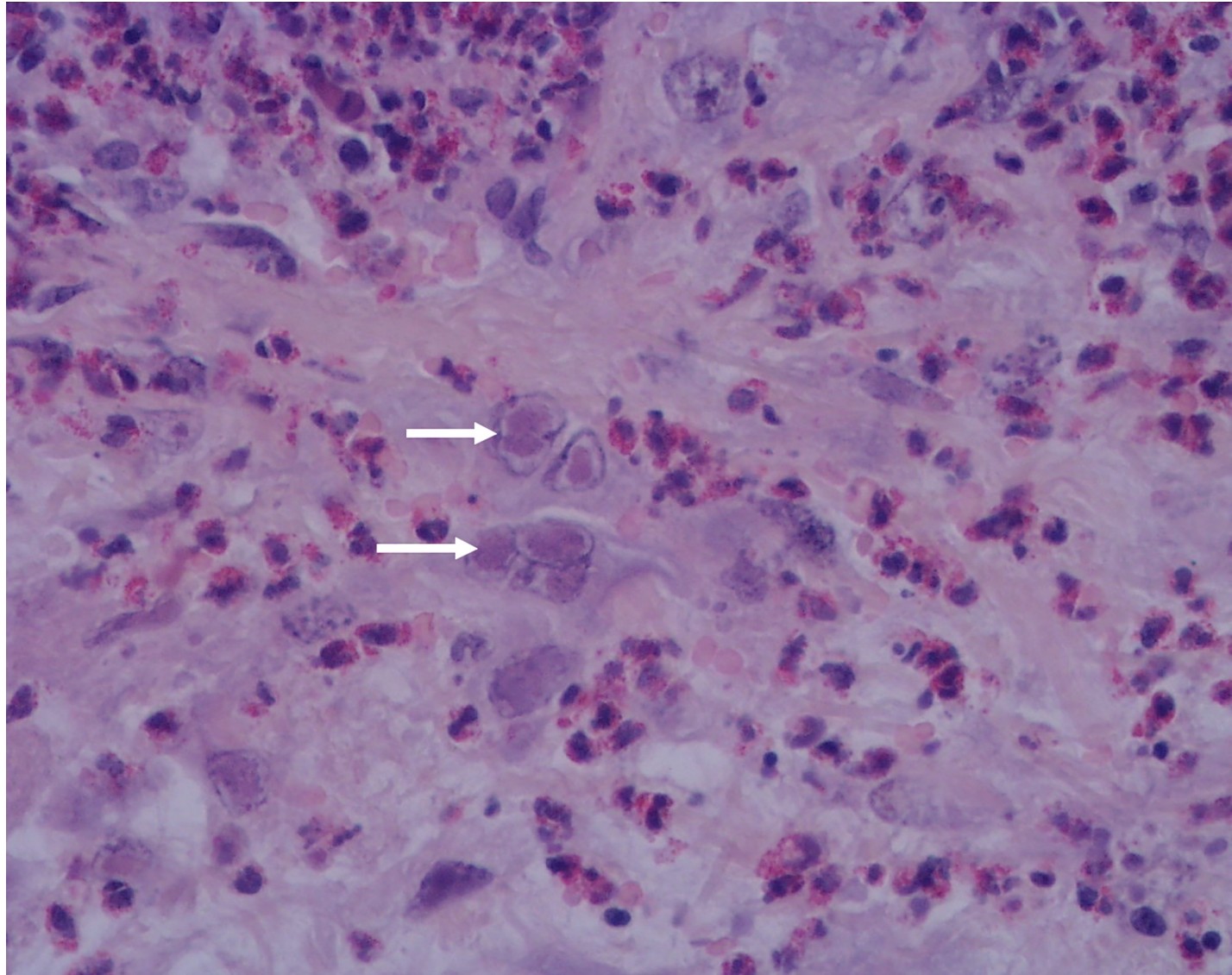

**Fig 4. Penis of hare-1.** Intranuclear inclusion bodies in mesenchymal cells (arrows) and moderate to severe infiltration by heterophils, H&E, 400x.

Six hares showed doubtful results from which four were MYXV-negative. Parasitological and bacteriological examinations did not reveal any infections that could justify the death of these animals.

## 3.5. Molecular characterisation of Iberian hare herpesvirus

For 50% of the animals (19/38), an amplicon ~225 bp-long, compatible with herpesvirus, was obtained in the pan-herpesvirus PCR [17]. For six hare samples only a weak band was generated, therefore were not considered for further analysis. The presence of herpesvirus was confirmed by sequencing analysis in 16 hares. For 11 of these amplicons, the nucleotide sequences obtained were independently edited to remove the primer targeting sequences and assembled. The consensus sequences (171 bp) showed 100% similarity to each other. Five of the obtained sequences were submitted to the GenBank (MN557129-33).

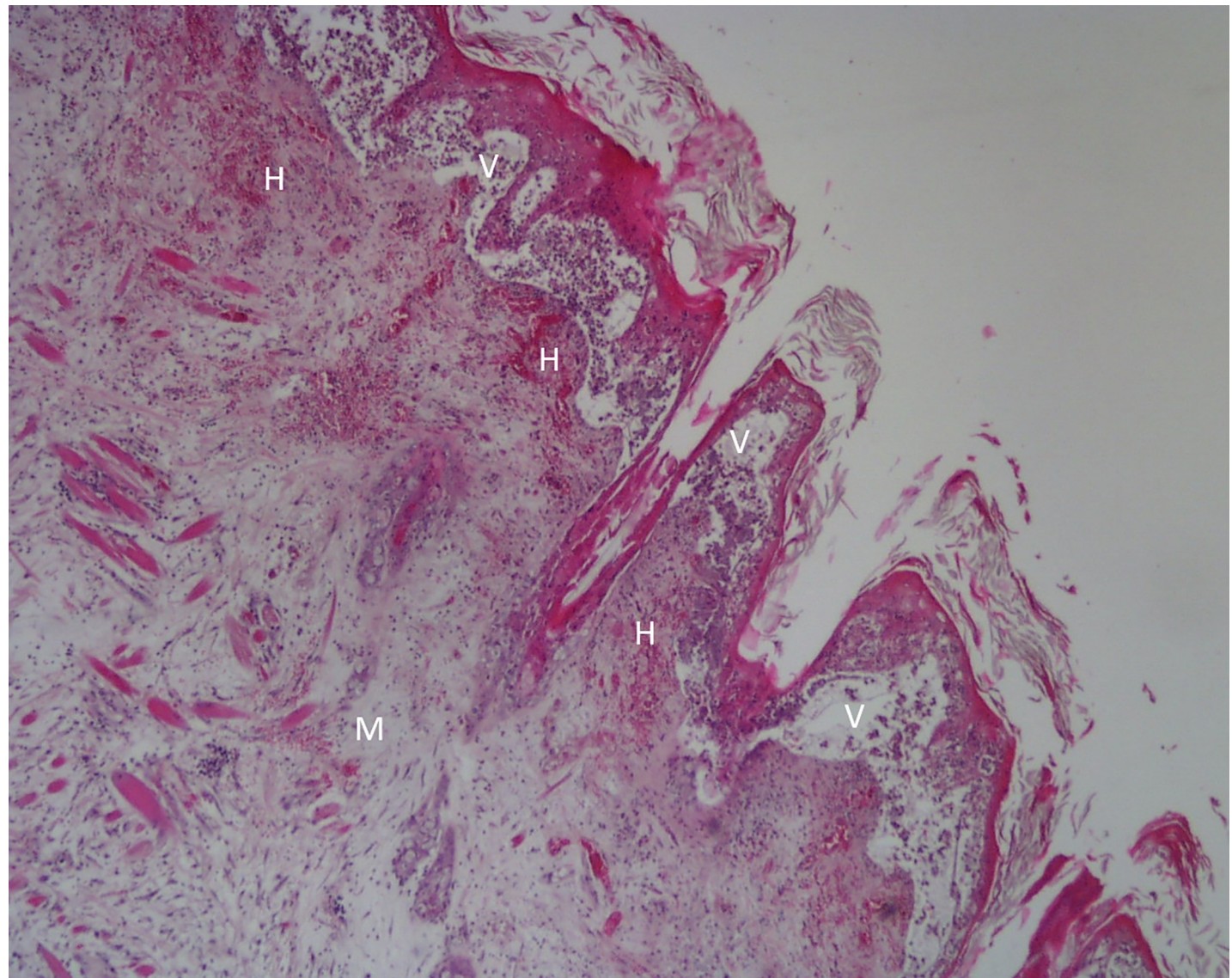

**Fig 5. Lip of hare-2.** Intraepidermal and subepidermal vesicopustules (V) that appear as "empty spaces" between the dermis and the epidermis, causing the detachment of the epidermis; mixomatous tissue (M) in the dermis characterized by abundant extracellular matrix pulling apart the fibroblasts; microhemorrhages (H) in the uppermost layer of the dermis;. H&E, 40x.

NCBI blast analysis (28.02.2020) of the DNA polymerase nucleotide sequences confirmed homology with herpesvirus *DNA polymerase* coding sequence from other mammals. Though with a low query cover of 48%, 78.31% of similarity was observed with a Phocid herpesvirus 2 (NC_043062.1) and 77.11% with Megabat gammaherpesvirus (LC268920.1) and a Harp seal herpesvirus (KP136799.1).

Blast analysis of the hares' DNA polymerase deduced aa sequences (28.02.2020), showed 52.63% identity over a query cover of 100% with bat herpesviruses (ALH21079.1, ALH21081.1 and ALH21071.1). Similarity was also found with Equid gammaherpesvirus 5 (AAD30141.1) showing 72.50% of identity and 62% of query cover, and with Asinine herpesvirus 4.1 (AAL14768.1), displaying 62% of identity over a query cover of 82%.

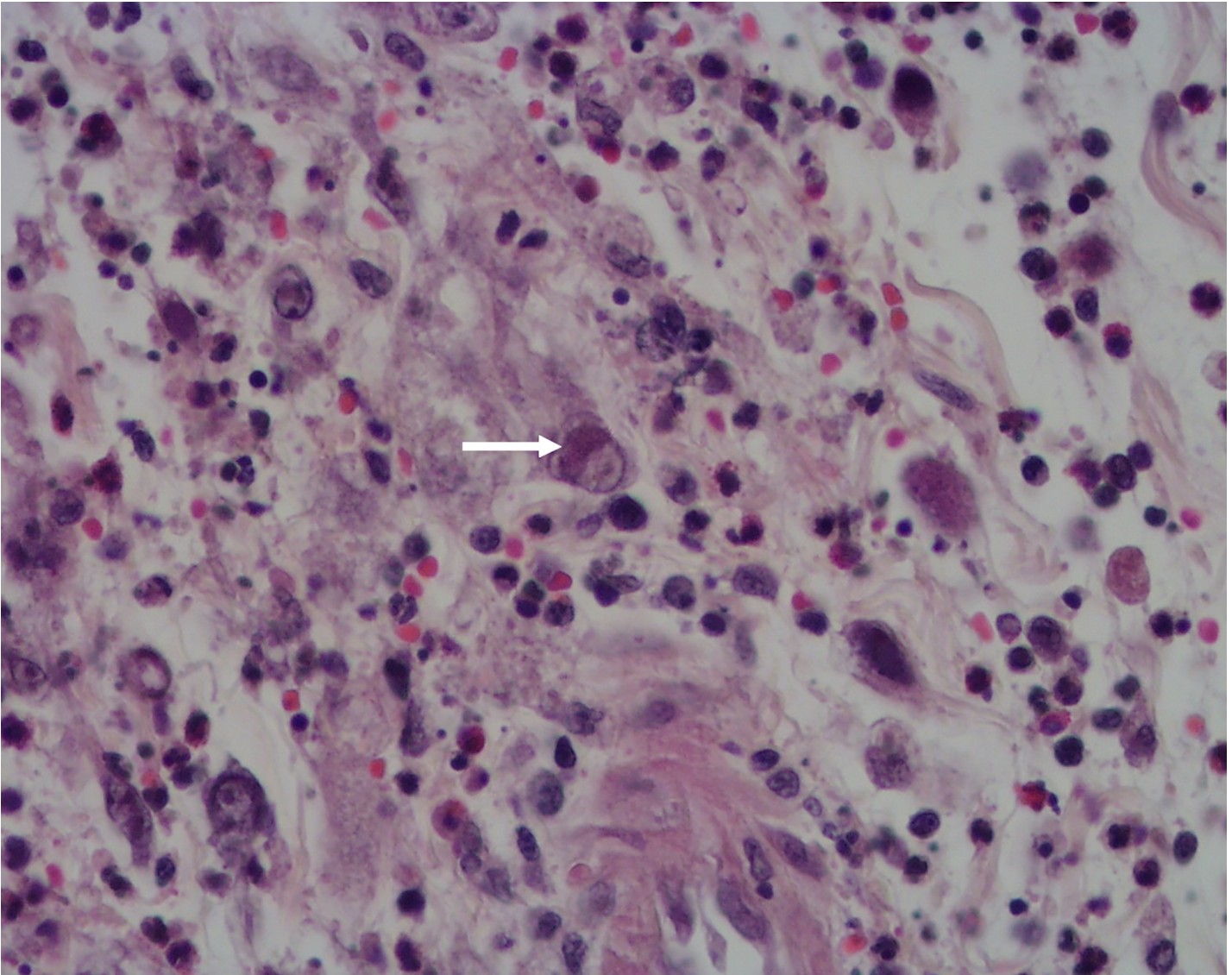

**Fig 6. Lip of hare-2.** Oedema and inflammatory cell infiltration with signs of necrosis. An intranuclear inclusion body in a mesenchymal cell can be seen (arrow). H&E, 400x.

An amplicon ~500bp long, was obtained from three DNA polymerase positive hare samples, with the genus-specific glycoprotein B (gB) gene primers described by [17].The three gB consensus sequences (453 bp long) showed 100% similarity to each other. These sequences were submitted to the GenBank (MN557129-31).

Blast analysis (23.03.2020) of the MN557129 sequence confirmed homology with Glycoprotein B sequence of known herpesvirus, namely with wood mouse (*Apodemus sylvaticus*) herpesvirus (GQ169129.1, EF495130.1 and EF128051.2, showing 67.69% to 70.72% similarity with a query cover of 88% to 92%), bank vole (*Myodes glareolus*) rhadinovirus (AY854169.2, 69.56% similarity and 92% query cover), field vole (*Microtus agrestis*) rhadinovirus (EF128052.1,, 67.69% similarity and 92% query cover), and chimpanzee (*Pan troglodytes*) rhadinovirus 1 and 2 (GQ995451.1 and EU085378.1, 65–67% similarity with 97–98% query cover).

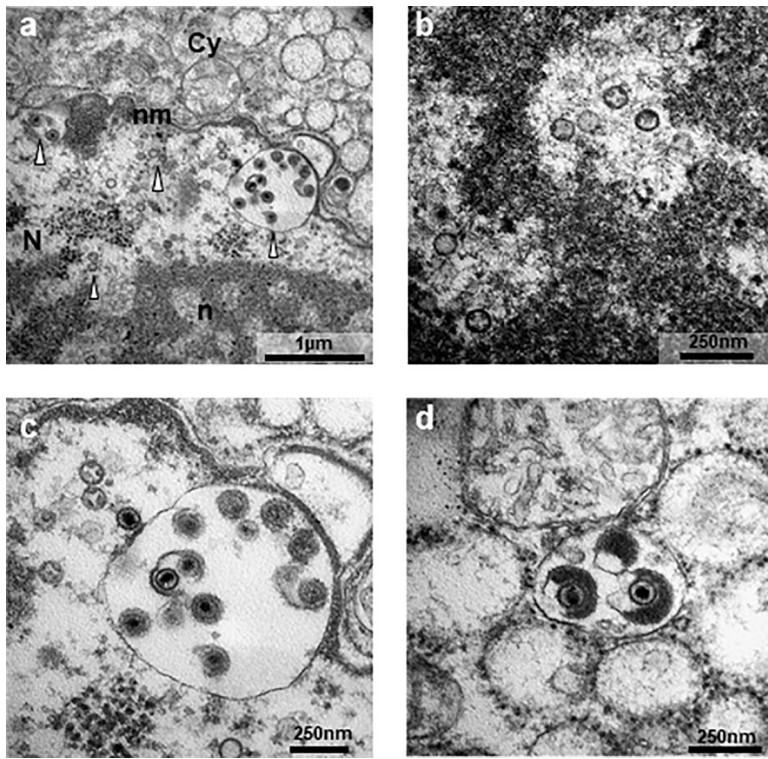

**Fig 7. Electron micrographs of penile soft tissue of hare-1. a-** Overview of a virus-infected cell (N, nucleus; n, nucleolus; nm, nuclear membrane; Cy, cytoplasm; arrowhead, viral particles); **b-** Naked capsids seen in areas of euchromatin in the nucleolus. **c-** DNA-loaded capsids close to the nuclear membrane in the process of budding into the perinuclear space; **d** Tegument assembly in the cytoplasm of the host cell. Photos obtained in a transmission electron microscope Hitachi H-7000 using iTEM software and Megaview III mid-mounted camera.

## 3.6. Phylogenetic analysis

It has been shown that, despite reduced, the region amplified by the nested PCR has discriminatory power to allow phylogenetic inferences [16]. To investigate the phylogenetic relationship of the Iberian hare herpesviruses found in this study (represented by sequence MN557129), with other members of the Herpesviridae family, a set of 37 DNA polymerase protein sequences from alpha-, beta- and gammaherpesviruses, obtained from GenBank, were edited to span a 54 to 59 aa-residue region comprising the homologous regions encoded by sequence MN557129.

Despite many polytomies and low bootstrap values, the analysis of the DNA polymerase protein sequences by unrooted Maximum Likelihood method and LG+G+I model [20] corroborated that the herpesvirus sequence from *Lepus granatensis* grouped within the gammaherpesvirus cluster (Fig 8).

To refine the phylogenetic inference within gammahespesviruses, the variability within the gB protein was explored in a set of 45 gammaherpesviruses using the clades described by [17] as reference. Two trees were constructed, the first based on the partial gB protein sequences alone, and the second based on concatenated DNA polymerase and gB sequences. Concatenation enabled greater phylogenetic resolution. The tree with the highest log likelihood (-3335,05) is shown in Fig 9. The accession numbers of the original sequences from which the DNA Polymerase and Glycoprotein B genes were edited, are indicated in the respective legend (Fig 9).

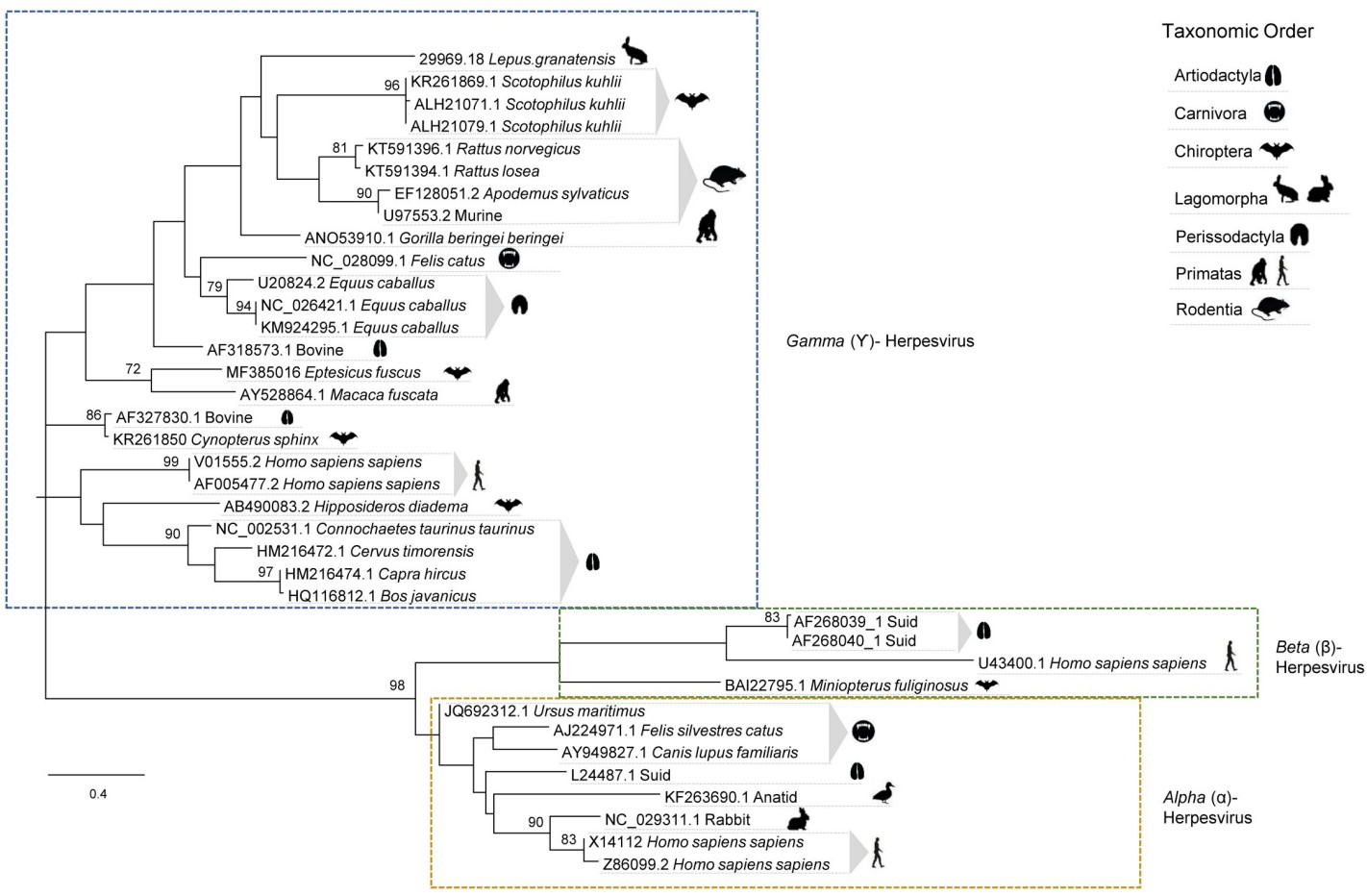

**Fig 8. Phylogenetic analysis based on 37 partial *DNA Polymerase* amino acid sequences of herpesviruses from several vertebrate species.** The access number of the nucleotide sequences from which the amino acid sequences were deduced are given. The tree with the highest log likelihood (-2432.55) is shown. The LG+G+I model considering 5 categories, [+G] parameter of 0,9929 and [+I] of 11,47% sites was used. The tree is drawn to scale, with branch lengths measured in the number of substitutions per site. There were a total of 60 positions in the final dataset. Robustness of the tree nodes was assessed by bootstrapping 1000 times. Only bootstrap values ≥70 are shown. The evolutionary analyses were conducted in MEGA X [18] and the phylogenetic tree was edited in the Figtree software version 1.4.0.

This phylogenetic analysis confirmed that the leporid herpesvirus under study is more closely related with gammaherpesviruses from Murine rhadinovirus (MHV-68 and MuHV-4), *Apodemus flavicollis* rhadinovirus (AflaRHV-1), *Apodemus sylvaticus* rhadinovirus 1 (AsylRHV1), *Apodemus sylvaticus* herpesvirus (WMHV), *Bandicota indica* rhadinovirus 4 (BindRHV-4), *Microtus agrestis* rhadinovirus 1 (MagrRHV1) and *Myodes glareolus* rhadinovirus 1 (MglaRHV1), but clearly diverge from this group, forming a separate clade supported by a bootstrap value of 100 (Fig 9). In this tree, no polytomies were observed.

Despite more information on the genome of this herpesvirus is required, this preliminary analysis suggest that it may represent a specific replicating lineage within the rhadinovirus genus. In accordance, we propose to name this virus Leporid gammaherpesvirus 5 (LeHV-5), following the rabbit alphaherpesvirus 4 (LeHV-4), the only leporid herpesvirus recognised so far as a species by the ICTV.

### 3.7. Isolation of the viruses in cell cultures

The difficulties found in viral isolation in CRFK, Vero, RK13 and Hella cells may be explained by the fact that LeHV-5 is a gammaherpesvirus.

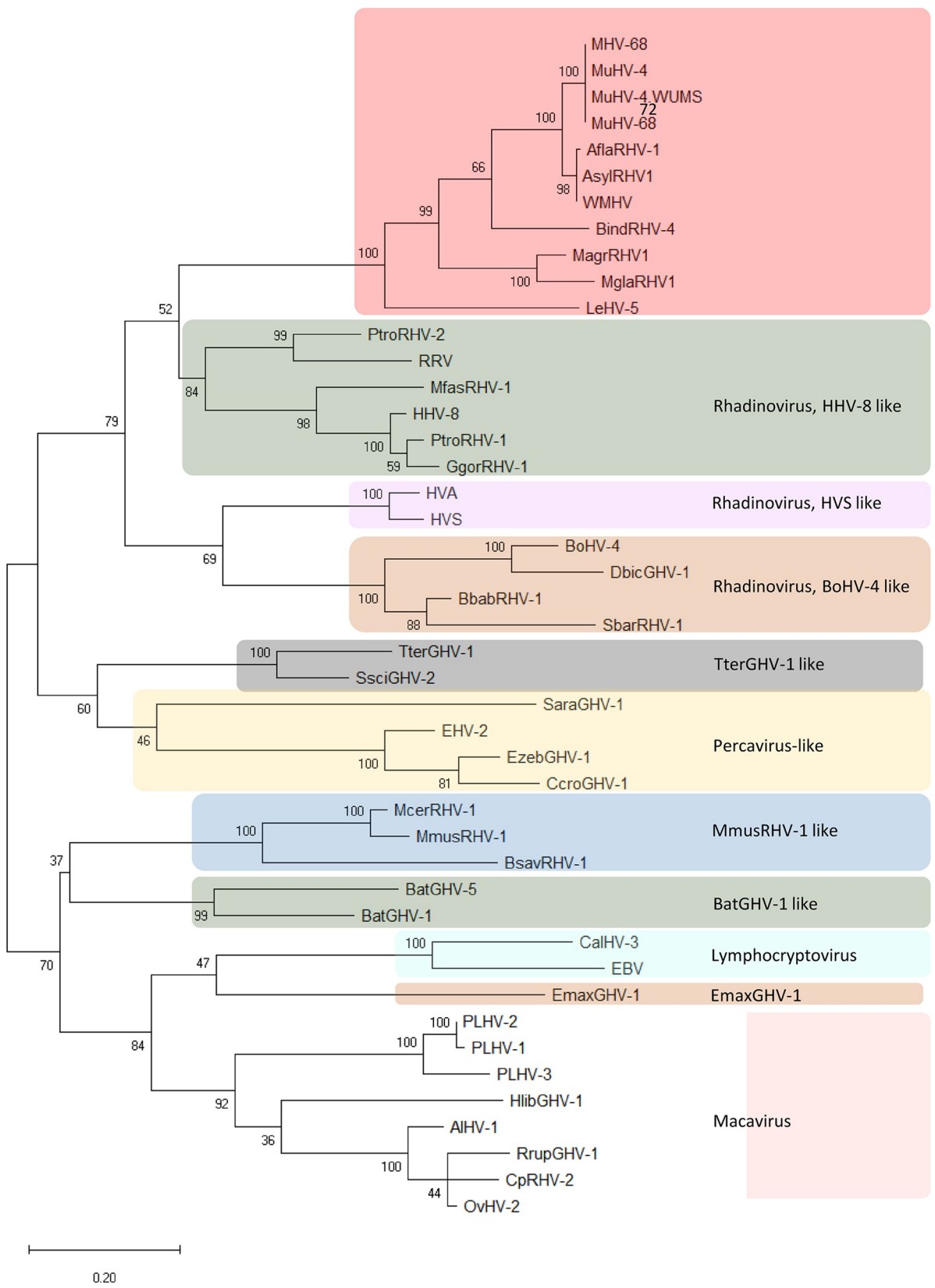

**Fig 9. Phylogenetic tree based on 45 concatenated DNA polymerase and glycoprotein B protein sequences (190 aa long) of several gammaherpesviruses inferred by using the Maximum Likelihood method.** The tree with the highest log likelihood (-6400.38) is shown. The L+G+I model [19] considering 5 categories, [+G] parameter of 0.85 and [+I] of 11.23% sites was used. There were a total of 188 positions in the final dataset. The tree is drawn to scale, with branch lengths measured in the number of substitutions per site. Robustness of the tree nodes was assessed by bootstrapping 1000 times. The evolutionary analyses were conducted in MEGA X [18] and the phylogenetic tree was edited in the Figtree software version 1.4.0. Accession numbers: MuHV-68 (U08990.1), MuHV-4 (AF324455.1), MuHV-4.WUMS (NC_001826.2), MuHV (DQ424896.1), AflaRHV-1 (DQ821580.2), AsylRHV1 (EF128051.2), WHMV (GQ169129.1), BindRHV-4 (DQ821581.1), MagrRHV1 (EF128052.1), MglaRHV1 (AY854169.2), PtroRHV-2 (EU085378.1), RRV (AF029302.1), MfasRHV-1 (AY138583), HHV-8 (U75698), PtroRHV-1 (AY138585.2), GgorRHV-1 (AY177144), HVA (AF083424), HVS (X64346), BoHV-4 (AF318573), DbicGHV-1 (AY197560), BbabRHV-1 (AY177146), SbarRHV-1 (AY177147), TterGHV-1 (AF141887), SsciGHV-2 (AY138584), SaraGHV-1 (EU085380), EHV-2 (NC 001650), EzebGHV-1 (AY495965), CcroGHV-1 (DQ789371), McerRHV-1 (DQ821582), MmusRHV-1 (AY854167), BsavRHV-1 (DQ821581), BatGHV-5 (DQ788629), BatGHV-1 (DQ788623), CalHV-3 (AF319782), EBV (AY037858), EmaxGHV-1 (EU085379), PLHV-2 (AY170317), PLHV-1 (AF478169), PLHV-3 (AY170316), HlibGHV-1m (AY197559), AlHV-1 (AF005370), RrupGHV-1 (DQ789369), CpRHV-2 (AF283477), OvHV-2 (NC_007646).

Despite LeHV-5 seems unable to multiply in human (Hela) and primate (Vero) cells, given the zoonotic potential of some animal herpesviruses, as the case of the cercopithecine alpha-herpesvirus 1 [21] and the murine gamma herpesvirus 68 [22], all isolation attempts were carried out in BSL-2 conditions.

## 4. Discussion and conclusion

This study describes the detection of the first herpesvirus (Leporid gammaherpesvirus 5) in the genus *Lepus* that, according to the phylogenetic analysis based on DNA polymerase and gB concatenated sequences, is more similar to rodent gammaherpesviruses of the rhadinovirus genus.

Viral isolation was not successful in RK13, CRFK or Vero cells. The difficulty in growing the virus *in vitro*, an important step towards its characterisation, may be a consequence of the co-infection with MYXV. The greater ease in the multiplication of MYXV, which grows in many cell cultures [23,24], may have hampered herpesvirus isolation. Moreover, wild animal samples are frequently somewhat autolysed and usually frozen before reaching the laboratory, which may lead to herpesvirus inactivation. The complexity in isolating genital gammaherpesvirus in cell culture was also referred by [25]. The lack of cell cultures from *Lepus* species may pose further challenges.

During our investigation, we observed necrosis of the genitals and herpetic-like vesicles in the lips of hares co-infected with LeHV-5 and MYXV, which were attributed to herpesvirus. The presence of LeHV-5 in the penile of hare-1 was confirmed by PCR and TEM, as shown in Fig 7. Herpesvirus particles were also visualised in epithelial and stroma cells of the eyelid of hare-2.

Notwithstanding the sampling limitation, the fact that macroscopic lesions were only seen in animals with myxomatosis, suggests that MYXV may play a role on herpesvirus replication and/or reactivation by compromising the immune response of the host, leading to the subsequent development of clinical disease with exuberant lesions. Immunosuppression facilitates herpesvirus infections and virus reactivation and it was demonstrated that certain MYXV proteins, such as Serp-1, have strong immune suppressing effects [24]. MYXV infection may hence represent a stress and/or immunosuppressive triggering factor for herpesvirus infection. It is known that stress, disease and other factors such as extreme temperatures can lead to the resurgence of herpesviruses in other species [26]. It was observed that depressed T-cell immune function reduces the quality of immune surveillance resulting in the increase of viral activity [27].

Herpesviruses generally follow one of three distinct strategies [27] within the host, namely i) latency with occasional re-emergence, ii) hit-and-run' approach and iii) slow-and-low' tactic

[27].In the case of LeHV-5, because it affects wildlife, specimens are mainly animals found dead or moribund, limiting the conclusions on the strategy of the virus.

In addition, herpesviruses are frequently found either in the absence of clinical signs or in association with very diverse clinical signs [28].This fact muddles the understanding of the true role and relative contribution of many herpesviruses in the courses of certain diseases, especially with regards to wild species that are often exposed to, and infected by, many pathogens. On the other hand, animal experimentation is complicated by the absence of available specific pathogen free (SPF) specimens, and by the difficulties in keeping hares in captivity, limiting cause-effect experiments. Moreover, during latency, herpesviruses may not be detected by current methods as it is the case of gammaherpesvirus in horses, resulting in an underestimated prevalence in the populations [28,'29]. Reports indicate that equine herpesvirus 2 (EHV-2) can be detected in immunocompetent animals in the absence of signs of disease (revised on [28]) meaning that healthy animals can be a potentially source of viral transmission.

According to our study, based on viral DNA amplification, around half of animals tested (63% symptomatic and 37% asymptomatic) were positive for LeHV-5. However, this value may be an underestimation given that the tropism of LeHV-5 is still unknown, and consequently the tissue samples used for diagnosis may have been inadequate. Thus, we cannot assure that PCR negative animals were not false negatives.

The fact that herpetic lesions were not observed in young, could mean that if the acquisition of LeHV-5 occurs at an early age, primary infection takes place with mild or no symptoms. In the absence of an unbalancing triggering factor such as MYXV infection, LeHV-5 may successfully establish a long-term relationship with the hare host, with subclinical disease and transient viremia. This would explain the detection of herpesvirus DNA in apparently healthy hares.

Interestingly, the gammaherpesvirus identified in external genitalia of the investigated hares was not associated with the development of papillary lesions as in other genital gammaherpesviruses' infections [25]. However, given that hunters, hunting managers and landowners have the opinion that the reproduction of the Iberian hare has been declining in recent years, it is important to clarify if this reduction is also associated with the emergence or circulation of LeHV-5. Other important concern is the potential production of oncogenic proteins by this herpesvirus.

According to our findings, genital herpesvirus may have a critical effect on hares' fertility and reproduction as well as in their survival. Hence, it is crucial to evaluate and understand the extent to which MYXV plays a role in the infection/reactivation of herpesvirus, as well as the putative role of herpesvirus in favouring infection of hares by MYXV or aggravating the severity of myxomatosis clinical forms.

The virological results obtained in this study also disclosed the infection of apparently healthy hares by LeHV-5, suggesting the possible circulation of this virus in the wild populations in a subclinical form. Because herpesvirus-DNA was detected in internal organs (liver and spleen), this asymptomatic infection may be systemic.

Although the Iberian hare populations are still considered stable, no census is available for the populations in Portugal. Presently, we continue monitoring apparently healthy and MYXV-positive hares in mainland Portugal to determine the extent of the geographic distribution of LeHV-5 among the wild hare populations, and the putative association of herpesviruses with the virulence of the recently emerged hare myxoma virus.

It is of paramount importance to evaluate the geographical distribution of the virus in the hare populations, the real extent and severity of the lesions induced in hares by LeHV-5, the persistence and latency of herpesvirus in the wild populations and the LEHV-5-MYXV

associated pathology in order to predict the consequences of the LEHV-5 infection at population level and to evaluate its importance in the future of this iconic species.

## Acknowledgments

We thank Sebastião Miguel (Hunting manager), João Carvalho (ANPC), Jacinto Amaro (FEN-CAÇA), Fernando Castanheira Pinto (CNCP) and Duarte Nuno (ICNF), for sample collection and organization of the sampling events. We are also grateful to Dr.Teresa Fagulha (INIAV, Virology Laboratory) for help with molecular characterization and to Maria João Teixeira (INIAV, Virology Laboratory) for technical assistance. Finally, we also thank to all the hunters who participated in fieldwork and sample collection.

## Author Contributions

**Conceptualization:** F. A. Abade dos Santos, M. D. Duarte.

**Data curation:** F. A. Abade dos Santos, C. L. Carvalho, M. D. Duarte.

**Formal analysis:** F. A. Abade dos Santos, C. L. Carvalho, M. D. Duarte.

**Funding acquisition:** F. A. Abade dos Santos, M. C. Peleteiro, M. D. Duarte.

**Investigation:** F. A. Abade dos Santos, M. D. Duarte.

**Methodology:** F. A. Abade dos Santos, A. Pinto, P. Carvalho, P. Mendonça, T. Carvalho, M. D. Duarte.

**Project administration:** F. A. Abade dos Santos, M. C. Peleteiro.

**Resources:** M. Monteiro.

**Software:** F. A. Abade dos Santos, C. L. Carvalho.

**Supervision:** F. A. Abade dos Santos.

**Validation:** F. A. Abade dos Santos, M. C. Peleteiro.

**Visualization:** F. A. Abade dos Santos.

**Writing – original draft:** F. A. Abade dos Santos, M. Monteiro, A. Pinto.

**Writing – review & editing:** F. A. Abade dos Santos, M. C. Peleteiro.

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
