## [Decision Letter · Decision Letter 0]

18 Feb 2020

PONE-D-20-00203

First description of a herpesvirus infection in genus Lepus

PLOS ONE

Dear Dr Santos,

Thank you for submitting your manuscript to PLOS ONE. After careful consideration, we feel that it has merit but does not fully meet PLOS ONE’s publication criteria as it currently stands. Therefore, we invite you to submit a revised version of the manuscript that addresses the points raised during the review process.

I strongly believe that revising the manuscript with suggested changes will further enhance the quality of manuscript. The reviewers concerns regarding the phylogenetic analysis should be considered. One of the reviewers submitted their comments before the missing figures could be provided to reviewer and hence the comments related to missing figures has been taken care by me.  

We would appreciate receiving your revised manuscript by Apr 03 2020 11:59PM. To enhance the reproducibility of your results, we recommend that if applicable you deposit your laboratory protocols in protocols.io, where a protocol can be assigned its own identifier (DOI) such that it can be cited independently in the future. For instructions see: http://journals.plos.org/plosone/s/submission-guidelines#loc-laboratory-protocols

We look forward to receiving your revised manuscript.

Kind regards,

Binod Kumar, PhD

Academic Editor

PLOS ONE

Journal Requirements:

2. In your Methods section, please give the sources of any cell lines used in your study.

3. In your Methods section, please provide additional location information, including geographic coordinates for the sampling locations/data set if available.

4. In your Methods section, please provide additional information regarding the permits you obtained for the work. Please ensure you have included the full name of the authority that approved the field site access and, if no permits were required, a brief statement explaining why.

6. Please include a copy of Table 2 which you refer to in your text on page 8-9.

Reviewers' comments:

Reviewer's Responses to Questions

**Comments to the Author**

1. Is the manuscript technically sound, and do the data support the conclusions?

Reviewer #1: Partly

Reviewer #2: Partly

2. Has the statistical analysis been performed appropriately and rigorously? 

Reviewer #1: Yes

Reviewer #2: No

3. Have the authors made all data underlying the findings in their manuscript fully available?

Reviewer #1: No

Reviewer #2: Yes

4. Is the manuscript presented in an intelligible fashion and written in standard English?

Reviewer #1: Yes

Reviewer #2: Yes

5. Review Comments to the Author

Reviewer #1: Review comments for the manuscript number: PONE-D-20-00203

In the present manuscript, authors have identified a new herpesvirus Leporid gammaherpesvirus 5 (LeHV-5), in genus Lepus. They confirmed the presence of herpesvirus in 13 MYXV-positive hares by PCR and sequencing analysis.

However, I would like to suggest authors to pay attention to following points:

1. Table 1 has not been numbered correctly.

2. Table 2 is missing

3. Figure legend must be little more descriptive

4. Figures 6, 7, 8 are missing

5. Bacteriological and parasitological examination method should have had been described in detail under methodology section.

6. Description of figure 2 under the heading Electron microscopy is confusing.

7. Results for epithelial and stroma cell of the eyelid are not shown but discussed. Either show the results or don’t discuss.

8. There are few spelling mistakes .

Reviewer #2: Dos Santos et al describe the detection of a yet unknown gammaherpesvirus in the Iberian hare by pathology, electron microscopy, PCR, sequencing, and phylogenetic analysis. The manuscript is of interest for virologists and readers interested in hares and their diseases. The manuscript is well structured, but contains a number of mistakes, misleading phrases which should be eliminated (see accompanying word document).

The main criticism pertains the phylogenetic analysis. Many branches of the trees are not statistically well supported. This is not surprising as the authors could only use the sequence that was derived from the generic PCR, i.e., appr. 175 bp, without the primer binding sequences (these have to be removed). It is important to perform phylogenetic analysis with extended sequences that result in statistically well supported trees. Methods to extend gammaherpesvirus sequences from the region of generic DPOL PCR into the coding sequence of glycoprotein B were published in a number of papers (e.g. Journal of Virology, 82(7), 3509-3516.). In the revised manuscript the authors should include statistically well supported trees.

6. PLOS authors have the option to publish the peer review history of their article (what does this mean?). If published, this will include your full peer review and any attached files.

Reviewer #1: No

Reviewer #2: No

---

## [Author Response · Author response to Decision Letter 0]

4 Mar 2020

Reviewer #1: Review comments for the manuscript number: PONE-D-20-00203 In the present manuscript, authors have identified a new herpesvirus Leporid gammaherpesvirus 5 (LeHV-5), in genus Lepus. They confirmed the presence of herpesvirus in 13 MYXV-positive hares by PCR and sequencing analysis.

However, I would like to suggest authors to pay attention to following

1. Table 1 has not been numbered correctly. 

The content and format of Table 1 was revised. LeHV-5 also included to allow comparing all the leporid herpesvirus described so far.

2. Table 2 is missing. 

Table 2 was included.

3. Figure legend must be little more descriptive 4. 

Legend was re-written.

Figures 6, 7, 8 are missing 5. 

All figures were included in this reviewed version of the manuscript.

Bacteriological and parasitological examination method should have had been described in detail under methodology section.

More detailed information was added to the manuscript as requested.

6. Description of figure 2 under the heading Electron microscopy is confusing.

Reference to Figure 2 was moved within the sentence.

7. Results for epithelial and stroma cell of the eyelid are not shown but discussed. Either show the results or don't discuss. 

At this time, the figure has not enough quality to be include. If Reviewer #1 agrees, we would like to remove this sentence from the Results section and mentioned this finding in the Discussion.

8. There are few spelling mistakes .

We thank Reviewer #2 for correcting these mistakes.

Reviewer #2: Dos Santos et al describe the detection of a yet unknown gammaherpesvirus in the Iberian hare by pathology, electron microscopy, PCR, sequencing, and phylogenetic analysis. The manuscript is of interest for virologists and readers interested in hares and their diseases. The manuscript is well structured, but contains a number of mistakes, misleading phrases which should be eliminated (see accompanying word document).

Thank you for the time revising the language and content of our manuscript.

The main criticism pertains the phylogenetic analysis. Many branches of the trees are not statistically well supported. This is not surprising as the authors could only use the sequence that was derived from the generic PCR, i.e., appr. 175 bp, without the primer binding sequences (these have to be removed). It is important to perform phylogenetic analysis with extended sequences that result in statistically well supported trees. Methods to extend gammaherpesvirus sequences from the region of generic DPOL PCR into the coding sequence of glycoprotein B were published in a number of papers (e.g. Journal of Virology, 82(7), 3509-3516.). In the revised manuscript the authors should include statistically well supported trees.

Thank you for point out this mistake. When preparing the alignments, the primer sequences were quite useful for the splicing, and we forgot to remove then.

As indicated, the 5’ and 3’ ends of the sequences corresponding to the primers, were removed and the trees reconstructed. As expected no significant phylogeny changes were observed.

We agree with Reviewer#2, that the region is quite small and that extending the plylogenetic analysis to a wider region would produce more robust data. However, most of the sequences available in the GenBank are also very short, and refer exactly to this region. In fact, given the high variability found in herpesviruses, the nested PCR developed by Van Devanter et al, is commonly used by many researchers, limiting the sequences available to the size of the second PCR-amplicon (≤ 220 nt long). That is the case of the sequences KR261864 and KR261869, both from Scotophilus kuhlii, the herpesviruses more closely related with LeHV-5, but also KT591396 from Rattus norvegicus, and U97553 from Murine.

Consequently, for the purpose of our phylogenetic analysis, extending the sequence would also reduce tremendously the number of herpesvirus hosts included. 

Also, as referred in the manuscript (Van Devanter et al, 1996), this small region has discriminatory power for phylogenetic inferences.

Responses to Reviewer #3

1. *** Sentence in yellow is misleading*** (Over the years, the Iberian hare has been unaffected by viral diseases that, alongside environmental and anthropogenic factors, led to the drastic decline of the wild rabbit). 

Sentence was rephrased to “Contrarily to the wild rabbit, which drastic decline has been linked, among other factors, to viral epizooties, until recently, the Iberian hare was not affected by viral diseases. Environmental and anthropogenic factors, however, have had a negative impact on both hare and wild-rabbit populations.”

2. (revised on ***???*** Jin et al.,2008) Of these, the most common naturally occurring herpesvirus infections identified in rabbits are LHV-2 and LHV-3, which alongside LHV-1 belong to the Gammaherpesvirinae subfamily.

We rewrote to “Of these, the most common naturally occurring herpesvirus infections identified in rabbits are LeHV-2 and LeHV-3 (reviewed by [9]), which alongside LeHV-1 belong to the Gammaherpesvirinae subfamily.”

3. ***MHV68 is used as a model virus since many years***( The lack of a suitable animal model …)

Thank you for pointing out this mistake. The sentence was erased. 

4. ***spell out*** (For histopathology, skin and genitalia fragments were fixated in 10% neutral buffered formalin, routinely paraffin embedded, sectioned at 4 μm, and stained with H&E)

Done

5. ***what is this?***(From these hunted specimens, no genitalia/skin samples were available for histopathology. Six hares showed doubtful results).

Sentence was changed to “Herpesvirus-DNA was also detected by PCR in the liver, spleen and lung samples of 41.2% (7/17) of the apparently healthy hunted hares that tested negative for MYXV. From this group of hares, no genitalia/skin samples were available for histopathology.”

6. ***The PCR products are of appr. 225 bp length. But after subtraction of the primer sequences, the novel sequence has a length of appr 175 bp only !*** (To refine the phylogenetic analysis with regards to gammahespesviruses (Figure 8), we explored the nucleotide variability among this group, using a second set of 25 herpesviruses sequences from orders Artiodactyla, Carnivora, Chiroptera, Lagomorpha, Perissodactyla, Primates and Rodentia. The accession numbers of the original sequences from which the DNA Polymerase 225 nt long sequences were edited, are indicated in Figure 8)

Thank you for point out this mistake. When preparing the alignments, the primer sequences were quite useful for the splicing, and we forgot to remove then.

As indicated, the 5’ and 3’ ends of the sequences corresponding to the primers, were removed and the trees reconstructed. As expected no significant phylogeny changes were observed.

7. ***None oft he LHVs is currrently classified as species by ICTV***( According with the most recent International Committee on Taxonomy of Viruses (ICTV) guidelines for classification of viruses, we propose to name this virus species leporid gammaherpesvirus 5 (LeHV-5), following the rabbit alphaherpesvirus 4 (LHV-4), although we cannot suggest a genus for LeHV-5). 

We rephrased the sentence. According to the 2018b Guidelines of ICTV, LHV-4 was included as a specie of an unclassified genus. Otherwise, LHV-1, LHV-2 and LHV-3 were not yet classified. 

8. ***??? Cercopithecine herpesvirus-1 is in the Alphaherpesvirinae! There is no zoonotic murine gammaherpesvirus***, (Despite LeHV-5 not appearing to multiply in human (Hela) and primate (Vero) cells, because gammaherpesviruses may be zoonotic, as is the case of cercopithecine herpesvirus-1 and murine gamma herpesvirus13 isolation attempts was performed under BSL-2 conditions.)

Thank you for pointing out this mistake. According with what is available in the literature, we rephrased the sentence to “ Despite LeHV-5 seems unable to multiply in human (Hela) and primate (Vero) cells, given the zoonotic potential of some animal herpesviruses, as the case of the cercopithecine alphaherpesvirus 1 [20] and the murine gamma herpesvirus 68 [21], all isolation attempts were carried out in BSL-2 conditions.”

9. *** Please rephrase *** (This study describes the detection of a new gammaherpesvirus in the genus Lepus that, according tophylogenetic analysis, is most similar to bat and rodent gammaherpesviruses.)

Done

10. ***for what?***(However, this value may be an underestimation given that the tropism of LeHV-5 is still unknown, and consequently the tissue samples used for diagnosis may have been inadequate) 

The sentence was rewritten

---

## [Decision Letter · Decision Letter 1]

23 Mar 2020

PONE-D-20-00203R1

First description of a herpesvirus infection in genus Lepus

PLOS ONE

Dear Dr Santos,

Thank you for submitting your revised manuscript to PLOS ONE. After careful consideration, we feel that it has merit but does not fully meet PLOS ONE’s publication criteria as it currently stands. Therefore, we invite you to submit a revised version of the manuscript that addresses the points raised during the review process.

You will notice that one of the reviewers is still not very satisfied with the quality of the phylogenetic trees. Please address this concern and resubmit the manuscript. 

We would appreciate receiving your revised manuscript by May 07 2020 11:59PM. To enhance the reproducibility of your results, we recommend that if applicable you deposit your laboratory protocols in protocols.io, where a protocol can be assigned its own identifier (DOI) such that it can be cited independently in the future. For instructions see: http://journals.plos.org/plosone/s/submission-guidelines#loc-laboratory-protocols

We look forward to receiving your revised manuscript.

Kind regards,

Binod Kumar, PhD

Academic Editor

PLOS ONE

Reviewers' comments:

Reviewer's Responses to Questions

**Comments to the Author**

1. If the authors have adequately addressed your comments raised in a previous round of review and you feel that this manuscript is now acceptable for publication, you may indicate that here to bypass the “Comments to the Author” section, enter your conflict of interest statement in the “Confidential to Editor” section, and submit your "Accept" recommendation.

Reviewer #1: All comments have been addressed

Reviewer #2: (No Response)

2. Is the manuscript technically sound, and do the data support the conclusions?

Reviewer #1: Yes

Reviewer #2: Partly

3. Has the statistical analysis been performed appropriately and rigorously? 

Reviewer #1: Yes

Reviewer #2: No

4. Have the authors made all data underlying the findings in their manuscript fully available?

Reviewer #1: Yes

Reviewer #2: Yes

5. Is the manuscript presented in an intelligible fashion and written in standard English?

Reviewer #1: Yes

Reviewer #2: Yes

6. Review Comments to the Author

Reviewer #1: Dos Santos et al detected a new gammaherpesvirus in the Iberian hare using pathological, microscopic, and phylogenetic analysis tools. In light of comments from reviewers, authors have done extensive revision and the current form of manuscript is suitable for acceptance for publication.

1. Authors have included all the figures that were missing in the original manuscript.

2. As suggested, the portion for epithelial and stroma cell of the eyelid has been removed from the result section. However, it can be discussed.

3. Bacteriological and parasitological examination has been explained in details.

Reviewer #2: I still criticize the quality of the phylogenetic trees, due to usage of extremely short sequences. Most parts of the gammaherpesvirus clades are without statistical significance. This is also highlighted by the fact that the sistervirus of the novel lepus HV is different in the 2 trees, either hipposideros HV or Scotophilus HV. All in all, the position of the lepus HV in the gammaherpesvirus clades is largely undefined. Therefore, these trees only show that the lepus HV is a gammaherpesvirus (which can be seen in simple BLAST analysis). So either, the tree analysis has to be improved with longer sequences, or the trees shouldn´t be shown at all.

7. PLOS authors have the option to publish the peer review history of their article (what does this mean?). If published, this will include your full peer review and any attached files.

Reviewer #1: No

Reviewer #2: No

---

## [Author Response · Author response to Decision Letter 1]

30 Mar 2020

Dear Editor

Dr. Binod Kumar

We were pleased to see that Reviewer#1 is satisfied with the corrections done in the second version of our manuscript. 

We would like to emphasise that at the main outcome of our manuscript is the demonstration that a potentially pathogenic gammaherpesvirus is circulating the Iberian hare, which we believe we have fully accomplished through microscopic, molecular and histopathological evidences. 

Regarding the characterisation of the new LeHV-5, besides sequencing and comparison to the HV presently known by BLAST analysis, we believe that presenting a phylogenetic analysis, even if preliminary, is important and relevant. 

Therefore, to respond to Reviewer #2 concerns, we reformulated the phylogenetic analysis by obtaining the nucleotide sequence of the gB gene from 3 LeHV-5 strains. This 453 bp sequence was used to construct an aa phylogenetic tree and concatenated with the respective DNA polymerase sequences to produce an aa base tree.

Our results showed that the LeHV-5 clearly grouped inside the MuHV-68 like Rhadinovirus, despite presenting a divergence of herpesviruses from rodents, showing a clear phylogenetic separation, which better the quality and relevance of our work. The absence of available sequences from other leporid gammaherpesviruses made it impossible to compare them.

We hope to have now reached the quality standards for publication and hope to see our manuscript published soon in the Plos One Journal.

Kind regards

Fábio Abade dos Santos

Dear colleague

We really appreciate your comments and we understand that it has forced us to significantly improve the quality of our work

Response to Reviewer (#2).

I still criticize the quality of the phylogenetic trees, due to usage of extremely short sequences. 

It is true that our analysis was based on a small region of viral DNA polymerase (171 bp). However, according to Van Devander described in the manuscript, this region harbours enough discriminatory power to differentiate alpha, beta and gammahespesvirus. 

Due to the situation of COVID-19, we have serious problems in investigative work. So we decided to include a partial Gb sequence, and present a phylogenetic study based on the concatenated Gb and DNA polymerase sequences. In this way, we understand that our analysis is already robust enough.

Most parts of the gammaherpesvirus clades are without statistical significance. 

It is also true that the bootstrap value for the LeHV-5 note in the DNA polymerase based phylogenetic tree was low. In accordance, the fragility of the analysis was clearly mentioned in the manuscript. At this moment, our clades are more significant and concordant with clades described previously by other manuscripts.

This is also highlighted by the fact that the sister virus of the novel lepus HV is different in the 2 trees, either hipposideros HV or Scotophilus HV.

This fact have changed with the new phylogenetic analysis.

All in all, the position of the lepus HV in the gammaherpesvirus clades is largely undefined. Therefore, these trees only show that the lepus HV is a gammaherpesvirus (which can be seen in simple BLAST analysis). 

We evaluated the similarity of the query sequence with the nucleotide sequences available in the GenBank at the moment by BLAST analysis. However, we believe that a phylogenetic inference tree based on the optimal global alignment of the all the sequences represented provides additional important information with regards to the evolutionary phylogeny of our group of sequences, meaning that one analysis does not substitute the other.

So either, the tree analysis has to be improved with longer sequences, or the trees shouldn’t be shown at all.

Instead of extending the DNA polymerase sequences, given the limitation referred to above, we constructed new phylogenetic trees following two strategies;

1. Using the gB gene sequences, which was amplified using the system described by Ehlers. 

2. Using concatenated DNA polymerase/gB sequences 

Kind Regards

Fábio Abade dos Santos

---

## [Editor Report · Decision Letter 2]

1 Apr 2020

First description of a herpesvirus infection in genus Lepus

PONE-D-20-00203R2

Dear Dr. Santos,

We are pleased to inform you that your manuscript has been judged scientifically suitable for publication and will be formally accepted for publication once it complies with all outstanding technical requirements.

With kind regards,

Binod Kumar, PhD

Academic Editor

PLOS ONE
---

## [Editor Report · Acceptance letter]

3 Apr 2020

PONE-D-20-00203R2 

First description of a herpesvirus infection in genus Lepus 

Dear Dr. Santos:

I am pleased to inform you that your manuscript has been deemed suitable for publication in PLOS ONE. Congratulations! Your manuscript is now with our production department. 

With kind regards,

on behalf of

Dr. Binod Kumar 

Academic Editor

PLOS ONE